# MgO Catalysts for FAME Synthesis Prepared Using PEG Surfactant during Precipitation and Calcination

**Valdis Kampars \*, Ruta Kampare and Aija Krumina**

Institute of Applied Chemistry, Faculty of Material Sciences and Applied Chemistry, Riga Technical University, LV-1048 Riga, Latvia; ruta.kampare@rtu.lv (R.K.); aija.krumina@rtu.lv (A.K.)
\* Correspondence: valdis.kampars@rtu.lv

**Abstract:** To develop a method for the preparation of MgO nanoparticles, precatalyst synthesis from magnesium nitrate with ammonia and calcination was performed in presence of PEG in air. Without PEG, the catalysts are inactive. The conversion to hydroxide was performed using a PEG/MgO molar ratio of 1, but, before the calcination, excess of PEG was either saved (PEG1) or increased to 2, 3, or 4 (PEG 2–4). Catalysts were calcined at 400–660 °C and characterized using XRD, $N_2$ adsorption-desorption, TGA, FTIR, and SEM. The FAME yield in the reactions with methanol depend on the PEG ratio used and the calcination temperature. The optimal calcination temperature and highest FAME yield in the 6 h reactions for catalysts PEG1, PEG2, PEG3 and PEG4 were 400 °C, 74%; 500 °C, 80%; 500 °C, 51% and 550 °C, 31%, respectively. The yield dependence on calcination temperature for catalysts with a constant PEG ratio is similar to that of a bell curve, which becomes wider and flatters with an increase in PEG ratio. For most catalysts, the FAME yield increases as the size of the crystallites decreases. The dependence of FAME and the intermediate yield on oil conversion confirms that all catalysts have strong base sites.

**Keywords:** heterogeneous catalysts; combustion; PEG; transesterification; biodiesel

## 1. Introduction

Biodiesel is one of the three main biofuels and is the main biofuel in the EU. The European biodiesel market has been stabilized at around 10 million tons/year [1]. Biodiesel is currently produced from high-quality vegetable oil (EU rapeseed oil) by transesterification with methanol at 65 °C in the presence of a homogeneous basic catalyst using a methanol/triglyceride molar ratio of 6/9. Reaction time does not exceed 1h. In parallel with the production of biofuels from high-quality edible oils, there is a growing trend towards the use of cheap and non-edible raw materials for the production of advanced biodiesel [2]. The RED II Directive provides for a gradual transition for Member States from conventional to advanced biofuels, with the following shares of energy consumption in the transport sector: 0.2% in 2022, 1% in 2025, and 3.5% in 2030 [3]. The production of advanced biofuels makes it possible to use local raw materials, including waste, and to reduce the extremely harmful effects of the transportation sector on the environment and climate.

Regardless of the raw material, the currently dominant acquisition process involves esterification in the presence of a homogeneous catalyst, and has various significant drawbacks:

- The process can be carried out if the triglyceride does not contain more than 1–2% free fatty acids (FFA);
- The separation of biodiesel from glycerol and glycerides is not fully realized;
- The quality of both biodiesel and the by-product glycerol is low and the treatment of crude products polluting the environment is necessary;
- The catalyst material is not reusable and enters the product and the purification system;

- The production of biodiesel from low-quality feedstock requires pre-treatment and the resulting raw material has increased the content of free fatty acids (FFA), so the use of homogeneous catalyst causes soap formation.

All the above-mentioned problems could be solved by using heterogeneous basic catalysts [4,5]. The development of these catalysts was intensified by the RED II Directive requirements, but the achieved results so far are unsatisfactory. Transesterification reactions in the presence of active basic heterogeneous catalysts usually take place at 90–150 °C for 4–8 h [6,7]. The only heterogeneous catalyst developed by the AXENS corporation, which has proven to be good and durable in industrial operations, works at a temperature of 190–220 °C and pressure of 40–70 bar [8]. As the currently dominant biodiesel production technology is implemented at atmospheric pressure, it is not expected that developments will take place in terms of the complexity of the production infrastructure and there is a significant increase in production cost by using high temperature and pressure. Undoubtedly, the most promising direction of development in the industry is processing at low temperature in the presence of more active and robust heterogeneous catalysts.

Biodiesel synthesis from triglyceride (TG) proceeds according to the balanced summary equation (Equation (1)):

$$TG + 3\,MeOH \rightleftharpoons 3\,FAME + G \tag{1}$$

where MeOH—methanol; FAME—mixture of fatty acid methyl esters (biodiesel); and G—glycerol.

Reaction (1) consists of three reversible stages and proceeds with the production of intermediates, such as diglycerides (DG) and monoglycerides (MG). In the presence of homogeneous basic catalysts, reaction is initiated by generation of methoxide anion (2), which attacks the carbonyl group of TG and causes fast FAME production without losing the active anion intermediates:

$$MeOH + B^- \rightleftharpoons MeO^- + BH \tag{2}$$

where $B^-$ are $HO^-$ or $RO^-$.

All reactions are reversible, therefore, the partitioning of glycerol in the form of separate layers is very important for shifting the equilibrium (1) to the right. In order for the reactions to proceed rapidly and selectively, a necessary minimum $B^-$ concentration must be maintained [9]. In the presence of basic heterogeneous catalysts, the mechanism is considered to be similar, but the role of the base is to create strong base sites on the surface of the catalysts. Similar to homogeneous catalysis, the reaction is initiated by methoxide anion generation and the activity of the heterogeneous catalyst should be determined by the density of the strong base sites on its surface [10–13]. Most of the successfully investigated metal oxide basic catalysts for transesterification reactions belong to the M(II)O group and are ionic. The surface of these catalysts is terminated by metal cations and oxide anions ($O^{2-}$), and contains different types of defects and environments (kinks, steps, terraces), which play a determining role in the catalytic phenomenon [14]. Successfully tested catalysts for biodiesel production are BeO, MgO, CaO, SrO, and BaO [15]. From these, calcium oxide is favored as it is highly active, cheap, and eco-friendly [16]. Unfortunately, this catalyst is leached during reaction and loses its activity in contact with air, it therefore cannot be considered as a prospective choice [15]. More stable and promising is MgO, which is used in medicine as a catalyst and an adsorbent [17,18]. The polarizing power of magnesium ion (3.9) is remarkably stronger than that of calcium ion (2.0), which would make the building of strong base sites more difficult than in case of CaO [19]. This is confirmed by the high dependence of MgO catalyst activity on manufacturing conditions [20]. Solution combustion synthesis of catalysts increases activity by employing metal salts as oxidants and different organic compounds as fuels in order to use the heat of the redox reaction [21–23]. The characteristic stages of catalyst synthesis according to this method are: sol solution, viscous gel, dried gel, and calcined product preparation.

The most active oxidants are nitrates, but their mixture with organic materials is explosive and there are less prospective industrial uses. To prevent the explosiveness proceeding the calcination stage, conversion from the nitrates to the hydroxides is widely used [17]. The present work is undertaken to investigate the synthesis of active MgO catalysts, and avoiding explosion during calcination by replacing strong oxidants with a weak oxidant, such as air.

## 2. Results and Discussion

### 2.1. Preparation of Catalysts and Their Characteristics

Magnesium salts, such as nitrate, sulfate, chloride, carbonate, acetate, and citrate, are most often used as feedstocks for the synthesis of MgO. Often, these salts are not used directly as a source of magnesium, but are converted into a magnesium hydroxide precatalyst. This is based on the considerations that low-temperature decomposition of magnesium hydroxide yields pure magnesium oxide. Thus, it is possible to perform calcination without harmful gases. The addition of different organic compounds as fuels before calcination prevents the formation of explosive mixtures. Thermal decomposition of magnesium hydroxide is well investigated. Its low decomposition temperature (300–400 °C) allows easy activation of the precatalyst [20,24]. As presented in Figure 1, the calcination of $Mg(OH)_2$ at 375 °C is sufficient for obtaining the polycrystalline cubic structure of MgO nanoparticles.

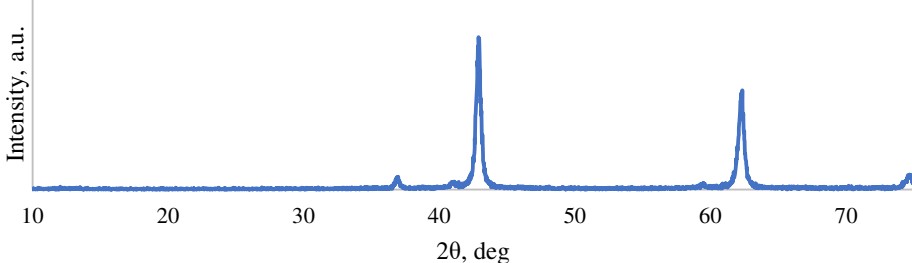

**Figure 1.** XRD patterns of PEG1, calcined at 375 °C for 3 h, obtained using Cu K-alpha radiation (0.15406 nm).

The peaks at 37°, 43°, 62.4°, and 74.7° correspond to (111), (200), (220), and (311) reflections, which reveal the formation of the polycrystalline structure of MgO nanoparticles [25]. No peaks from impurities were detected in the registered patterns. The sharp diffraction peaks indicate the good crystallinity of the MgO products.

It is known that higher activity of catalysts is obtained when fuel and surfactants are present in both precatalyst synthesis and in the activation stages [26–28]. During the preparation of a precatalyst, PEG 6000 works as a surfactant, but during the calcination it is as a surfactant and fuel. Our research shows that the presence of PEG during the synthesis of the precatalyst in a molar ratio of MgO of 1 creates a small effect, but a further increase in PEG shows no improvement up to 4. Therefore, in all cases, we performed the first step using a molar ratio of PEG 6000 to MgO of 1. The presence of surfactant and fuel at the calcination stage changes the structure and activity of the resulting catalyst and the effect depends on the ratio to catalyst [28]. Thermal analysis up to 700 °C (Figure 2) of precatalysts was performed to examine PEG 6000 effects in an inert atmosphere.

From the TG curves, it can be seen that samples PEG1 and PEG4 display a very notable mass loss near 400 °C, which is related to the decomposition of PEG 6000. In the row PEG1–PEG2–PEG3–PEG4, gradual changes in the TG curves occur, suggesting that the calcination temperature at 400 °C would be sufficient to obtain active MgO nanoparticles.

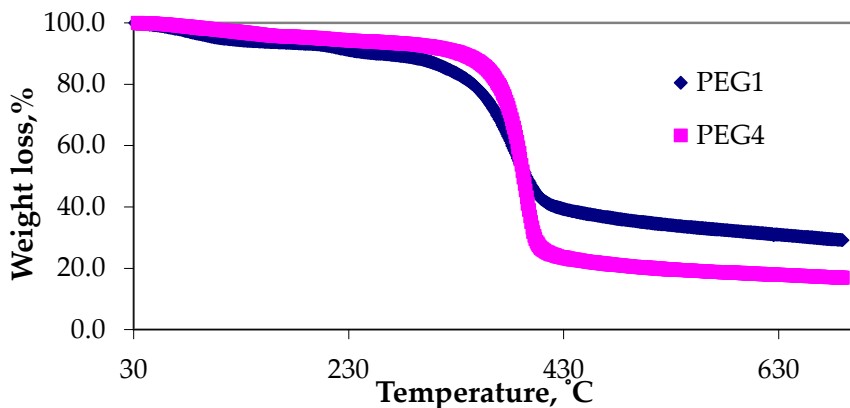

**Figure 2.** Thermogravimetric analysis of precursors of PEG1 and PEG4.

As shown in Figure 2, a slight weight loss is still observed in the temperature range of 430 °C to 730 °C, which can be explained by the elimination of adsorbed carbonate species [20].

The nitrogen adsorption–desorption isotherms and specific surface area, total pore volume, the pore sizes of the catalysts are shown in Figure 3 and Table 1, respectively. All catalysts have large pores with an average pore diameter of around 5 nm. As seen from Figure 3, the isotherms of MgO are type IV with an H3 hysteresis loop.

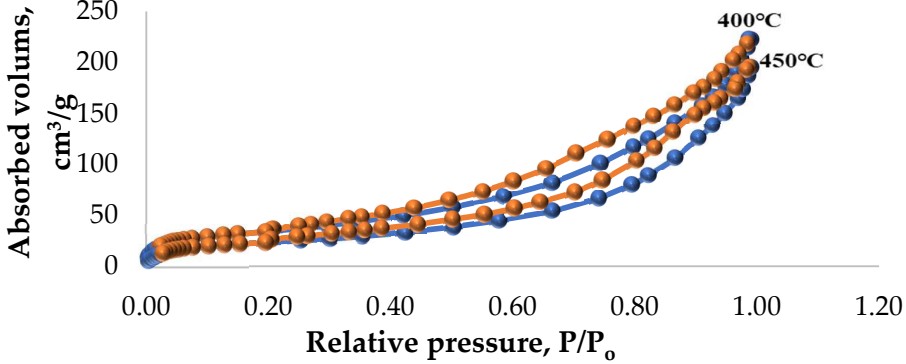

**Figure 3.** Nitrogen adsorption–desorption isotherms of PEG2 calcined at 400 °C and 450 °C.

**Table 1.** Textural properties of MgO.

| Catalyst | Total Surface Area, m²/g | Pore Volume, mL/g | Pore Size, nm | FAME Yield, % |
|---|---|---|---|---|
| PEG2 400 | 126.2 | 0.366 | 4.95 | 42.7 |
| PEG2 450 | 85.7 | 0.319 | 4.98 | 73.9 |

As seen in Table 1, the change in the textural characteristics is small and the number of performed experiments does not allow to draw conclusions about the effect of texture on the activity of the catalysts.

Characteristic SEM images of the obtained MgO are presented in Figure 4. Two different forms of aggregates dominate. These images indicate that the powder sample is an agglomeration of nanoparticles. Irregular aggregated flake-like particles with a rough surface represented the active catalysts, while MgO composed of many irregular nanoparticles did not catalyze the interesterification reactions. A similar phenomenon, with a significant morphological effect on the activity of catalysts or adsorbents, has also been observed in other works [29,30]. The aggregation process with the formation of a unique morphology may play a crucial role in ensuring catalytic activity.

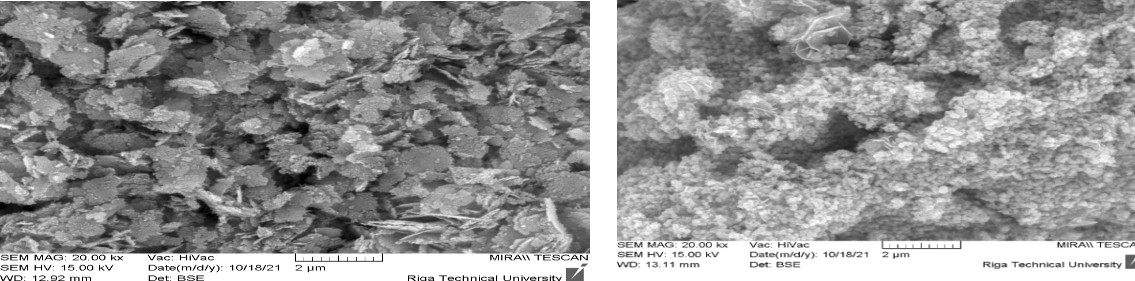

**Figure 4.** SEM image of active (**left**) and inactive (**right**) MgO catalysts.

### 2.2. Catalytic Performance

Initialization of the transesterification reaction proceeded according to Equation (2). The catalytic activity of heterogeneous catalysts depends on the density of strong base sites on the catalyst surface. This depends on the morphology, number, and types of defects [10–13]. The presence of PEG improved MgO activity. Catalysts synthesized without PEG under experimental conditions did not provide a FAME yield above 3%.

#### 2.2.1. Effects of PEG Content and Calcination Temperature

To determine the effect of PEG content and calcination temperature on catalytic performance, each PEG1–PEG4 precatalyst was calcined at a different temperature. As shown in Figure 5, sharp maxima of the FAME yield were registered for the PEG1 catalyst at 400 °C, for the PEG2 catalyst at 500 °C, for the PEG3 catalyst at 500 °C, and there was a flatter maximum for PEG4 at 550 °C.

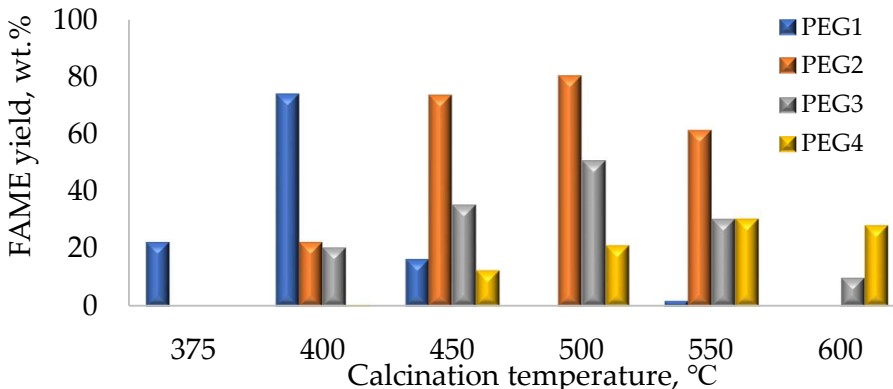

**Figure 5.** Effect of PEG content and calcination temperature on FAME yield in transesterification reactions at 65 °C after 6h.

The repetition of individual points showed that the standard deviations of the FAME yield of the PEG1 catalyst at 400 °C reached 15.5%. It also showed that the density of strong base catalytic sites depends on more than just the calcination temperature. It was also affected by minor changes in the calcination process, including air circulation, crucible form, the volume of precatalyst in the crucible, and location in the muffle furnace. As the combustion process proceeds by the reaction of fuel with oxygen in the air, this seems understandable and requires a more precise process for providing airflow control. The standard deviation of the results was lowered by increasing the PEG content, but a PEG content above 3 lowered the activity. The shape of the distribution of the experimental points from the calcination temperature for the catalyst with a constant PEG ratio is similar to a bell curve and becomes wider and lower with an increase in PEG content. PEG2 provides the best catalysts.

The dependence of the activity of MgO catalysts on small changes in the manufacturing process is well known and the safe production of an active MgO catalyst for practical use is indeed a serious problem. Babak et al. concluded that unsupported MgO was not a suitable

catalyst, but Li et al. showed that mesoporous-supported MgO catalyst only becomes active at 220 °C [31,32]. At the same time, works on very high activity MgO catalysts have been published [6,26,27]. The problem of MgO catalyst activity has been analyzed using specific surface properties in the form of research and quantum mechanics calculation methods. Di Cosimo et al. confirmed that the surface of the magnesium oxide catalyst obtained by the calcination of magnesium hydroxide contains three types of sites with different basicity: surface sites of strong (low coordination $O^{2-}$ anions), medium (oxygen in $Mg^{2+}$-$O^{2-}$ pairs), and weak (OH- groups) basicity [20]. Decomposition of $Mg(OH)_2$ at 400 °C generates hydroxylated MgO, mainly containing strong $O^{2-}$ basic sites located in surface defects, such as the corners and edges of the crystalline solid surface. An increase in the calcination temperature removes the OH groups and also surface solid defects, creating more stable structures that contains a higher concentration of medium-strength $Mg^{2+}$-$O^{2-}$ basic pair sites. Thus, an increase in the calcination temperature drastically decreased the density of strong base sites and, to a lesser extent, that of weak OH groups, while it slightly increased that of medium-strength base sites; therefore, a calcination temperature of 400 °C seems to be optimal [20]. According to Montero et al., water and methanol chemisorb preferentially over defects and edge sites over NanoMgO-700 through the conversion of surface $O^{2-}$ sites to $OH^-$ and the coincident creation of Mg-OH or $Mg-OCH_3$ moieties, respectively [33]. According to this work, the best calcination temperature for magnesium hydroxide would be 700 °C [33]. It could be considered that both publications indicate the area fir the best calcination temperatures and the results for PEG catalysts, which narrow this area to 400–600 °C (Figure 5).

According to Vedrine, the main role in creating of strong base sites is different for different types of defects [14]. If this is so, then reducing the size of the crystallites should increase the density of the defects and also catalyst activity. As follows from Figure 6, in general, the FAME yield and, thus, the density of strong base sites, increase as the size of the crystallites decreases. This could indicate an association between the defects of the boundary surfaces of these crystallites with the number of strong basic centers.

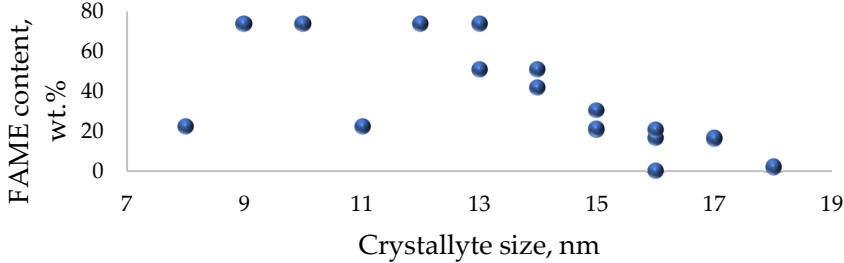

**Figure 6.** Relationship between FAME yield and crystallite size. The main crystallite size is calculated using (200) and (220) reflections using the Scherrer equation (Table S1).

However, two catalysts fall out of this relationship (PEG1, FAME content 22.5 wt.% and PEG2, FAME content 73.9 wt.%), which confirms that the size of the crystallites does not always determine the density of strong base sites, and other phenomena can rule out these relationships. An unexpected decrease in catalyst activity could be caused by partial blocking of the MgO surface with combustion products, or products resulting from incomplete calcination. To determine if this was the case, FTIR spectra were recorded for individual catalysts. The obtained FTIR spectra confirmed that not all catalysts have a clean MgO surface (Figure 7), although XRD and SEM-EDS analyses did not indicate this. Absorption bands near 1420 are assignable to magnesium carbonate or hydromagnesite [34], which is not uncommon for oxide catalysts. Therefore, catalyst surfaces control, determined with FTIR, should be included in determining the characteristics [35].

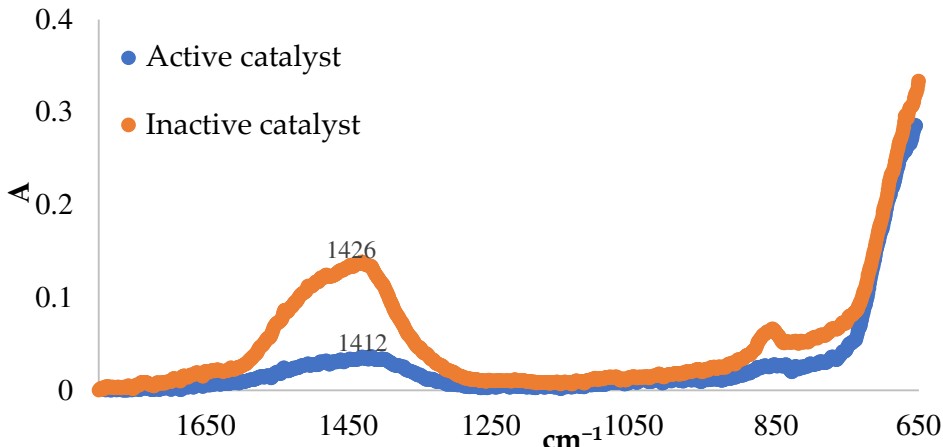

**Figure 7.** FTIR spectra of active (PEG2 crystallite size 13 nm, FAME 73.5%) and inactive (PEG1 crystallite size 8 nm, FAME 22.5%) catalysts.

2.2.2. Strong Base Sites and Reaction Proceeding

In reality, reaction (1) proceeds in three reversible stages:

$$TG + MeOH \rightleftharpoons FAME + DG \tag{3}$$

where DG are diglycerides,

$$DG + MeOH \rightleftharpoons FAME + MG \tag{4}$$

where MG are monoglycerides, and

$$MG + MeOH \rightleftharpoons FAME + G \tag{5}$$

where G is glycerol.

Chromatographic analysis shows that the main constituents of the obtained reaction mixtures were FAME, MG, DG, and TG, if the amount of G did not exceed 1 wt.%. The sum of these components was usually 97–100%, which allowed us to consider that the amount of unknown by-product was insignificant. Assuming that the reaction mechanism was provided by the same active sites in all cases, the change in the composition of the reaction products depending on oil conversion should reflect the gradual realization of the transesterification. As follows from Figure 8, the experimental points form logical curves that confirm the presence of the same active sites in all cases.

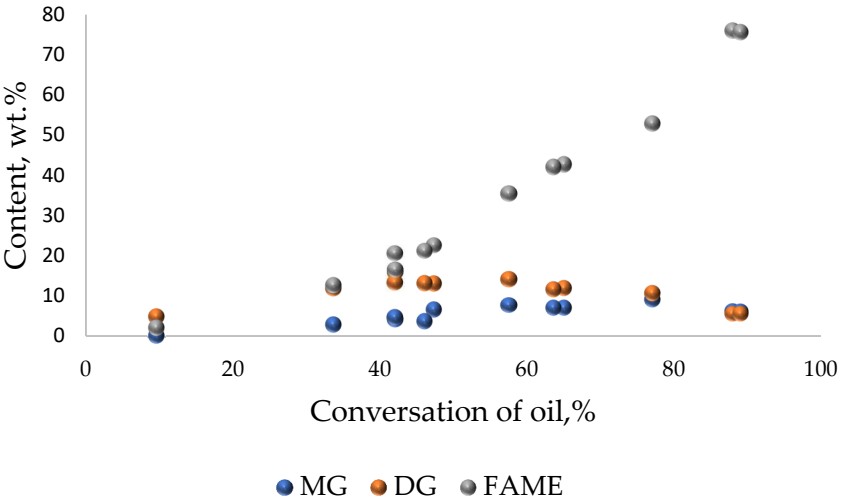

**Figure 8.** Composition of reaction mixture as a function on the conversion of oil.

The contents of MG and DG go through a maximum and approach zero above 95% conversion. DG content reaches a maximum with oil conversion of about 55%. However, the maximum content of MG was observed at an oil conversion of 70%. Up to 40% oil conversion, FAME content increases relatively slowly. As oil conversion increases, FAME content grows faster. In the conversion region, 40–80% FAME content increases at least twice as efficient than below 40%. Most likely, this is due to changes in the reaction rate of the reverse reactions caused by the gradual release of the glycerol in a separate layer.

## 3. Materials and Methods

### 3.1. Materials

Refined rapeseed oil was purchased from a local producer Iecavnieks & Co Ltd (Iecava, Latvia). The average molecular weight of the oil was 896 g/mol, density–0.92 g/mL at 20 °C, saponification value–186.7 mg KOH/g, and acid value–0.12 mg KOH/g. The percentages of fatty acids in the oil were: palmitic–4.1%, stearic–1.4%, oleic–62.5%, linoleic–21.7%, linolenic–8.7%, arachidic–0.4%, and other fatty acids–1.2 wt.%. The content of MG (monoglycerides) was 0.0 wt.%, DG (diglycerides)–0.4 wt.%, and TG–99.6 wt.%.

Magnesium nitrate hexahydrate, ammonia solution 25%, PEG 6000 and methanol were supplied by Sigma-Aldrich (St. Louis, MO, USA). Materials for GC analysis—1,2,4-butanetriol (96%) and MSTFA (*N*-Methyl-*N*-(trimethylsilyl)trifluoroacetamide (97%)–were purchased from Alfa Aesar (Haverhill, MA, USA), tricaprin (98%)—from TCI Europe (2070 Zwijndrecht, Belgium), pyridine (99.5%)—from Lach-Ner (Neratovice, Czech Republic), and dichloromethane (99.5%) was supplied by Chempur (Karlsruhe, Germany). Grade 1Whatman filter paper was supplied by Sigma-Aldrich.

### 3.2. Catalyst and Biodiesel Synthesis

Catalyst synthesis: 100 mL of 1M magnesium nitrate solution (25.6 g/100 mL) was added to a 250 mL Erlenmeyer flask and 4.4 g of PEG 6000 was then dissolved. The obtained mixture was stirred at 500 rpm for 1 h (homogenization stage N1). Then, it was filtrated with a $NH_4OH$ (25% $NH_3$ basis) solution in a burette to adjust the pH up to 10. The $NH_4OH$ solution was dripped from the burette slowly in a drop-wise manner. After reaching pH 10, the second homogenization process continued for 2h by stirring (homogenization stage N2) and then hydrothermal treatment was performed by leaving the solution overnight. The material obtained after centrifugation was suspended in 100 mL of PEG 6000 solution (PEG1 4.4 g/100 mL; PEG2 8.8 g/100 mL; PEG3 13.2 g/100 mL and PEG4 17.6 g/100 mL) using an ultrasound environment. Water was slowly removed using a rotary evaporator. The obtained material was placed in oven at 80 °C for 5 h to remove excess water moisture. The dried sample was placed in a crucible, which was placed inside a muffle furnace, for calcination at the selected temperature for 3 h. The temperature of the furnace was increased at the very slow rate of $1\ °C \cdot min^{-1}$ until the desired temperature of calcination was attained. Fine amorphous MgO nanoparticles were prepared and collected.

Biodiesel synthesis: 0.64 g of MgO catalyst (7% by weight of oil) was added to a 100 mL round bottom flask containing 11 mL of methanol (methanol to oil molar ratio of 27). The mixture was allowed to reflux at 65 °C for 0.5 h at 600 rpm, using a hotplate with a magnetic stirrer. Then, 10 mL of oil (10.3 mmol) diluted with 10 mL of co-solvent THF was added and refluxed at 65 °C at 600 rpm for 6 h. The obtained product was allowed to settle for 3 h. The crude biodiesel was separated from glycerol using a separating funnel. FAME was filtered using Whatman filter paper and, after removing the THF and excess methanol, stored in a glass container for analysis.

### 3.3. Catalyst Characterization

The samples were out-gassed at 150 °C for 24 h before measurements. The total surface area was estimated using the Brunauer–Emmett–Teller (BET) method. Pore diameters were derived from desorption isotherms using the Barrett–Joyner–Halenda (BJH) method.

XRD analysis was performed using an X-ray diffractometer (Bruker AXS D8 AD-VANCE, Billerica, MA, USA) in the range of 10–75° (2θ) with Cu K-alpha radiation (0.15406 nm) and a step size of 0.02°.

The surfaces of the catalysts were studied with a Tescan MIRA3 LMU (Tescan, Brno, Czech Republic) scanning electron microscope. Prior to analysis, samples were coated with a gold layer using an Emitech K550X sputter coater.

Universal attenuated total reflectance-Fourier transform infrared spectroscopy (UATR-FTIR) was used for surface control of the catalysts. Measurements were carried out on a *PerkinElmer Spectrum 100* spectrometer connected to a *Universal ATR Sampling Accessory*. A spectral range 650–4000 cm$^{-1}$ was selected.

TGA analysis was performed using a thermogravimetric analyzer (PerkinElmer STA 6000). The heating rate was 10 °C/min and experiments were performed under a pure nitrogen flow of 20 mL/min, supervised using a mass-flow controller.

### 3.4. Biodiesel Characterization

Analysis of all components from each sample of raw biodiesel was carried out by using an Analytical Controls (Rotterdam, The Netherlands) biodiesel analyser, based on the Agilent Technologies (Santa Clara, CA, USA) gas chromatograph 7890A. All compounds were analyzed using DB5-HT column (15 m, 0.32 mm, 0.10 μm) under conditions similar to those prescribed by standard EN 14105. Glycerol (G), monoglicerides (MG), diglicerides (DG), triglycerides (TG), and FAME were quantified as in our previous report [36]. The oven temperature was initially set to 50 °C for 5 min. Then, the temperature was first increased to 180 °C at a rate of 15 °C/min, then, to 230 °C at a rate of 7 °C/min, and, finally, to 370 °C at a rate of 10 °C/min. Helium was used as a carrier gas and the detector temperature was set to 390 °C. The reaction mixture was characterized by the mass percent content of the mentioned groups of compounds. The standard deviation for all analyzed compounds was not higher than 0.5 wt.%, but FAME and TG were not higher than 0.9 wt.%.

All experiments were repeated triplicate in order to determine the variability of the results and to assess experimental errors in GC analysis. Arithmetic averages and standard deviations were calculated for all results. Statistical analyses were performed using Microsoft Excel (2013). The yield of FAME was assumed to be equal to the wt.% of FAME content in the reaction mixture after removing glycerol, methanol, and the co-solvent.

### 4. Conclusions

From the alkaline earth metal oxides, MgO can be considered as the most promising catalyst for biodiesel synthesis. The chemical bond in MgO is less ionic than that in CaO, which makes it more stable but poses problems in the formation of a high density of strong base sites on the surface of nanoparticles to ensure a high catalytic activity. It is possible to significantly increase the activity of MgO nanoparticles using PEG as a surfactant and fuel, as well as with an optimal calcination temperature. FAME yield dependence on calcination temperature for catalysts obtained in presence of a constant molar PEG/MgO ratio is like a bell curve, the width, height, and position of which depend on that molar ratio. Increasing the PEG/MgO ratio has an optimum catalyst activity at 2. With the presence of catalysts PEG1, PEG2, PEG3, and PEG4, the maximum yield of FAME was achieved as 400 °C, 74%; 500 °C, 80%; 500 °C, 51%, and 550 °C, 31%, respectively. The narrowest zone of high activity was for the PEG1 catalyst and the widest and flattest was for PEG4. The PEG2 catalyst remaining the most prospective.

For most of the catalysts, the FAME yield increased as the size of the crystallites of catalyst decreased. The use of FTIR spectra showed that deviations in this relationship may be due to the retention of incomplete calcination products on the surface of MgO. FAME and intermediate yield dependence on oil conversion confirmed that all catalysts had the same types of strong base sites that were necessary for initialization of transesterification reactions.

**Supplementary Materials:** The following supporting information can be downloaded at: https://www.mdpi.com/article/10.3390/catal12020226/s1, Table S1. Reaction products by using different catalysts; Figure S1. XRD analysis of PEG1 catalyst Calcined at 400 °C; Figure S2. XRD analysis of PEG2 catalyst Calcined at 400 °C.

**Author Contributions:** Conceptualization, funding and final approval V.K.; methodology V.K., R.K. and A.K.; original draft preparation, editing V.K. and R.K., performed experiments and data collection R.K. and A.K. All authors have read and agreed to the published version of the manuscript.

**Funding:** This work was conducted as the project of the Latvian Council of Science Izp-2020/2-0194.

**Data Availability Statement:** Data are contained within the article and the Supplementary Materials.

**Conflicts of Interest:** The authors declare no conflict of interest.

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
