# Peer review of "MgO Catalysts for FAME Synthesis Prepared Using PEG Surfactant during Precipitation and Calcination"

_catalysts, doi:10.3390/catal12020226_

Round 1

Reviewer 1 Report

Kampars et al. describe the FAME biodiesel production using solid MgO base catalysts. The problem with MgO is that it is difficult to prepare with high surface area and the authors use the technique of adding polyethylene glycol (PEG) as a surfactant in the precipitation of Mg(OH)2 and as fuel during calcination to MgO.

The study is interesting and relevant for the catalysts journal, but needs major changes and more data before it is complete for publication. In addition, some language revision is needed.

First of all, the title and first sentence in the abstract does not make sense, how is PEG in the air? PEG 6000 is a solid. I could suggest a title like “MgO Catalysts for FAME synthesis prepared using PEG Surfactant during Precipitation and Calcination”

In order for the study to be complete all the prepared catalysts should be characterized with N2 physisorption and XRD to determine specific surface area, pore volume, average pore size and crystallite size. These are standard techniques and it should be possible to measure all samples and then compare to activity. Also, since it is a pure oxide the particle diameter can be estimated from the BET surface area assuming spherical particles as d = 6 / (SSA * density). For MgO the density is 3.58 g/cm³ and then for the PEG2 400 catalyst the particle diameter d = 6 / (126.2 m2/g * 3.58 * 106 g/m3) = 13.3 * 10-9 m = 13 nm. So, this fits well with the XRD determined particle size. This should be done for all samples. Raw data can be presented in the SI, with summary table/figure in the manuscript.

Figure 1 needs to be improved and state the X-ray wavelength or emission line (Cu K_alpha).  

Figure 3: Use different colors for the adsorption and desorption isotherm.

Figure 5 needs to be significantly improved, e.g. using a scatter plot and a proportional x-axis. The current x-axis is by category and several of the temperatures appear twice.

Figure 6: I would suggest to reverse the x- and y-axis, to have the obtained FAME content as function of the crystallite size.

Figure 7: In the caption note exactly which catalysts were measured.

Figure 8: What is the purpose of the polymeric fit? I suggest to delete it, as it is not derived from the reaction mechanism described in reactions (3) to (5).

References to the SI (the figures and table) seems to be missing in the main manuscript. And in the SI the figure captions should be placed below the figures.

In order to back up the main conclusion in the manuscript, that the activity is correlated to the density of base sites, there is missing an actual measurement of surface basicity, e.g. using CO2 temperature programmed desorption or IR spectroscopy with and acidic probe molecule (e.g. CO2 or MeOH).  In the introduction on lines 82-83 the authors also point out the need for determining the density of base sites.

Please revise the English language, as there are several places where the meaning is not clear, e.g. the title as described above, lines 9, 11-12, 60, 101, 362-363

Author Response

Dear Reviewer 1

Thank you for your objective assessment and many useful recommendations. I tried to follow them and the article really got better.

  • First of all, I would like to change the title of the article to what you suggested, because my original wording is not clear and unambiguous enough. The will be “MgO Catalysts for FAME synthesis prepared using PEG Surfactant during Precipitation and Calcination”.

  • N2 absorption-desorption measurements fits well with the XRD determined particle size. This should be done for all samples. Raw data can be presented in the SI, with summary table/figure in the manuscript.

I agree that N2 absorption-desorption measurements should have been performed on all the same samples that were subjected to XRD analysis. Unfortunately, we had serious problems with the provision of N2 absorption-desorption measurements and we were unable to develop this direction. As there is a good correlation between N2 absorption-desorption and XRD results, we have not lost the relationship between activity and crystallite size. This is the only problem I can't really fix, I'm sorry about.

  • Figure 1has been improved.
  • Figure 3 has been improved.
  • Figure 5 has been improved.
  • Figure 6 has been improved.
  • Figure 7 has been improved.
  • Figure 8 has been improved.
  • References to the SI (the figures and table) in the main manuscript have been tested.
  • The English language has been revised.

I agree that the value of the publication would have been significantly higher if we had carried out basic site studies on the surface of the catalysts. Unfortunately, the project in which the study was conducted was small and one-year. We failed to expand our research. We have planned research on the surface structure of catalysts in cooperation with the RTU Institute of Surface and Nanoobject Spectroscopy.

Thank you for your great recommendations and support.

Kind regards,

Valdis Kampars

Reviewer 2 Report

The publication presents a less dangerous way of synthesizing MgO, which was used as a catalyst for the synthesis of FAME.
There are several typos problems that need to be corrected:
- In lines 39-44 and 126 the indents and jumps in the text must be corrected.
- In line 5 the word "Inastitute" must be corrected
- Most of the graphics should be improved in quality. Especially necessary for figures 3, 5, 6, 7, 8. I recommend that they can have a similar appearance to figure 2. Even the type and color of the words in the graphics is different.
- Table 1 has another font.
- Figure 7 does not have the data label in orange.
- I recommend completing or renaming the label in figure 8 related to the polynomial curve (Poly. (FAME)).
- Check references 9 and 25. There are formatting problems.
- Review the abbreviation "ml" throughout the text. It should be mL.
- In line 280 add percentage (%) to the numbers.
- Review the separation of numbers and their units. For example, line 309 shows 6h and not "6 h". There are more cases like this in the text.
- What type of Whatmann filter paper was used in filtering FAME?
- Add the brand and model of the equipment used.

Regarding the discussion of results:

- Between lines 236-245 the authors argue that the size of the MgO crystals does not seem to be directly related to the density of stronge base sites. They emphasize, based on the results presented in figure 7, that the MgO nanoparticles seem to be covered on their surface with MgCO3 or with hydromagnesite: why did they not observe signals of these compounds in the XRD analyzes? In the SEM analysis did you do EDS analysis? It is possible that an EDS analysis also helped to verify or not the presence of MgCO3 or hydromagnesite. On the other hand, based on these same observations, if the authors declare that the correlation between MgO size and density of stronge base sites does not seem to be valid, however some lines below propose that this is not actually fulfilled due to the presence of MgCO3 or hydromagnesite, can you rule out this relationship between the size of the MgO nanoparticles and density of stronge base sites? Apparently it is not fulfilled just because the MgO nanoparticles are not pure.

Author Response

Dear Reviewer 2,

Thank you for valuable suggestions that I tried to use to improve the quality of this article. I started with Reviewer 1, I have already changed the title of the article and all the figures along with the captions. Therefore, I may not be able to make all your suggestions, but I hope that the realised revisions and improved English have made the article better.

Corrections made accordingly your recommendations:

  • Lines 39-44 indents and jumps….

Is done.

  • Line5 “Inastitute”.

Adjusted.

  • Revision of figures 3,5,6,7,8 …….

All figures are revised.

  • Font of Table1 …..

Adjusted.

  • Figure 7 ….

Is revised.

  • Figure 8 ….

Polynomial curve is removed.

  • References 9 and 25 ….

Are formatted.

  • Review the abbreviation “ml” …..

Has been reviewed.

  • Lne 280 add percentage (%) to the numbers.

Is added.

  • Review the separations of numbers and their units.

Is done.

  • What type of Whatman paper was used?

Whatman qualitative filter paper, Grade 1 from Sigma Aldrich.

  • Add a brand and model of the equipment used.

Is added.

  • Discussion between lines 236-245 …

Discussion has been clarified. We are convinced that such relationships exist, but it can be upset by various side effects, one of which is the surface purity of MgO. Obviously, after the calcination process, it is useful record the FTIR spectrum before the labor intensive further studies.

  • Conclusions are revised

Thank you again for your evaluation of the work and your valuable recommendations.

Valdis Kampars

Round 2

Reviewer 1 Report

The authors have improved the paper as much as possible according to my suggestions, and I recommend to publish after minor revisions to the Supplementary information:

Add the XRD estimated crystallite size for each sample and the one missing sum (PEG3 at 450 C) in Table S1. Place the captions of the figures below the figures.  

Author Response

Dear Reviewer,

I have perfected the Supplementary material according to your recommendations. Thanks again for your advice, which allowed me to significantly praise the quality of the article.

Valdis Kampars